# A Review: Natural and Synthetic Compounds Targeting *Entamoeba histolytica* and Its Biological Membrane

**DOI:** 10.3390/membranes12040396

**Published:** 2022-04-01

**Authors:** Nurhana Jasni, Syazwan Saidin, Norsyahida Arifin, Daruliza Kernain Azman, Lai Ngit Shin, Nurulhasanah Othman

**Affiliations:** 1Institute for Research in Molecular Medicine (INFORMM), Universiti Sains Malaysia, Gelugor 11800, Malaysia; hanajasni@yahoo.com (N.J.); syahida_arifin@usm.my (N.A.); daruliza@usm.my (D.K.A.); laingitshin@usm.my (L.N.S.); 2Department of Biology, Faculty of Science and Mathematics, Universiti Pendidikan Sultan Idris, Tanjung Malim 35900, Malaysia; syazwan@fsmt.upsi.edu.my

**Keywords:** *Entamoeba histolytica*, membrane, cytosolic proteins, compounds

## Abstract

Amoebiasis is the third most common parasitic cause of morbidity and mortality, particularly in countries with poor hygienic settings in central and south America, Africa, and India. This disease is caused by a protozoan parasite, namely *Entamoeba histolytica*, which infects approximately 50 million people worldwide, resulting in 70,000 deaths every year. Since the 1960s, *E. histolytica* infection has been successfully treated with metronidazole. However, there are drawbacks to metronidazole therapy: the side effects, duration of treatment, and need for additional drugs to prevent transmission. Previous interdisciplinary studies, including biophysics, bioinformatics, chemistry, and, more recently, lipidomics studies, have increased biomembranes’ publicity. The biological membranes are comprised of a mixture of membrane and cytosolic proteins. They work hand in hand mainly at the membrane part. They act as dedicated platforms for a whole range of cellular processes, such as cell proliferation, adhesion, migration, and intracellular trafficking, thus are appealing targets for drug treatment. Therefore, this review aims to observe the updated trend of the research regarding the biological membranes of *E. histolytica* from 2015 to 2021, which may help further research regarding the drug targeting the biological membrane.

## 1. Introduction

Amoebiasis is the third most common parasitic cause of morbidity and mortality, particularly in countries with poor hygienic settings in central and south America, Africa, and India [1,2]. This disease is caused by *Entamoeba histolytica*, a protozoan parasite that infects approximately 50 million people worldwide, resulting in an estimated 70,000 deaths every year [3]. This parasite has two major life cycle stages: the trophozoite and the cyst. The most common infection route is ingesting contaminated food and water, where the infection is concentrated in the intestine and organs such as the liver, lungs, and brain. While 90% of infected individuals are asymptomatic, the infection may also lead to severe complications, such as colitis with bloody diarrhea, liver abscesses, and colonic perforation [4].

Metronidazole is the most often prescribed and successful drug to treat *E. histolytica.* The modes of action involve four steps: passive diffusion into the cell membrane, reduction of nitro groups to nitro radicals by ferredoxin or flavodoxin (intracellular transport protein), generation of toxic metabolites, and finally the reaction of the metabolites with DNA and formation of adducts with guanosine [5,6]. This mode of action of metronidazole results in the *E. histolytica* death. However, there are drawbacks to metronidazole therapy such as its adverse side effects, qualities of being carcinogenic and mutagenic, alcohol intolerance, and problems when used during pregnancy and lactation [7,8]. Furthermore, the increasing concern of parasites developing resistance is also one of the drawbacks of this drug [7]. Therefore, identifying and characterizing specific targets is urgently needed to design new therapeutics to improve treatment against amoebiasis.

Biological membranes are a successful evolutionary result of a long-term natural selection process and one of the most complex structures that allow life to exist [9,10]. Biological membranes have various functions but the main function is to act as a key platform for the entire network of cellular processes, such as for cell proliferation, adhesion, migration, and intracellular trafficking, thus being appealing targets for drug treatment [11]. Apart from that, the phospholipids that are part of biological membranes are the necessary components for fusing vesicles and fundamental for the phagocytosis of *E. histolytica*. According to Niesen et al. (2017) [12] membrane proteins, including transporters, channels, and receptors, are the target; almost 70% of transporters are FDA-approved drug targets.

In addition, Azmi and Othman (2021) [13] reported the abundance of cytosolic proteins found in the membrane fractions in their research. Cytosolic proteins were mainly involved in protein modification, metabolic processes, signal transduction, mRNA degradation, and cell death. Their research also provides the predictive functions of the biological membranes and cytosolic protein. In addition to that, they also demonstrated the occurrence of interactions between them. This scenario proves that the membrane and cytosolic proteins play essential roles in pathogenesis, which helps in further searching for a suitable drug target.

To date, little is known about the molecules involved in this process, with only about 20 proteins or protein families found exposed to the *E. histolytica* surface and barely any information about the lipid [14].The molecules involved are illustrated in Figure 1. Despite this, there are still reports regarding the molecules as potential targets for developing amoebicidal drug candidates. Thus, this review aims to observe the trend of the research regarding the biological membranes, which consist of a mixture of membrane and cytosolic proteins of *E. histolytica*, which may help further research regarding the drug targeting these molecules.

## 2. Biological Membranes’ Protein

### 2.1. Thioredoxin Reductase

Thioredoxin reductase (TrxR) is one of the enzymes found in the surface proteome of *E. histolytica*. It is considered a membrane enzyme containing selenocysteine, a rare amino acid [15]. The presence of this amino acid makes it a selenoprotein. The observation was confirmed using western blotting and immunofluorescence microscopy [16]. The function of the TrxR is to catalyze the reversible transfer of reducing equivalents between reduced nicotinamide adenine dinucleotide phosphate (NADPH) and thioredoxin, a small protein that performs critical metabolic processes in maintaining the intracellular redox balance [17]. TrxR is often paired with thioredoxin, forming a thioredoxin system, a ubiquitous oxidoreductase system with antioxidant and redox regulatory roles [18].

The systems take part in several activities such as the regulation of enzymatic activities; repairing of oxidized proteins; affording of reducing equivalents for DNA synthesis; and cellular transcription, growth, and apoptosis [17,19,20]. This enzyme is a potential drug target because it exhibits pharmacokinetics, an availability of a three-dimensional structure, favorable physiochemical properties, participation in significant pathways, rich interactions, and broad-spectrum conservation [21]. Parsonage et al. (2016) [22] demonstrated a drug called auranofin that can target the TrxR. A possible mechanism of auranofin activity is that the monovalent gold, Au(I), released from the drug can attach to the thioredoxin reductase’s redox-active dithiol group (TrxR). Subsequently, the parasite’s redox equilibrium is predicted to be disrupted by Au(I) binding in the active site, preventing electron transport to the downstream substrate thioredoxin (Trx). However, the results of the experiments reported that auranofin is not suitable to target the thioredoxin reductase because the CXXC-catalytic motif has insufficient space for binding Au(I) by both cysteines of the dithiol group.

A compound (1-(carboxymethyl)-4-(4-methylthiazole-5-carboxamido)-3H-pyrazol-1-ium-3-ide) was virtually screened against the TrxR via molecular docking [21]. The result shows that the residues of the Glu33 and Ala115 atoms showed high affinity (hydrogen bonds) for inhibitors, though a minor tilting behavior was observed when hydrogen bond analysis was conducted. Screening of this antibacterial compound also presented the possibility of targeting the TrxR. In a current study conducted on *Homalomena aromatica* Schott, which is a plant used in different ethnomedicinal practices of Southeast Asia to treat stomach ailments against intestinal parasites, the authors found that three molecules, namely 3,7-dimethylocta-1,6-dien-3-yl acetate, α-methyl-α-(4-methyl-3-pentenyl)-oriranemethanol, and 7-octadiene-2,6-diol-2,6-dimethyl, were predicted to be potential leads against the TrxR [23].

### 2.2. Cysteine Protease

The following enzyme found on the membrane of *E. histolytica* is a cysteine protease, which is critically implicated in the pathogenesis of protozoic infections [13]. It contains the Cys–His–Asn triad at the active site. According to Tusar et al. (2021) [24] this enzyme uses the reactive site cysteine as a catalytic nucleophile and histidine to hydrolyze the peptide bond. These parasite-derived cysteine proteases also play critical roles in hemoglobin hydrolysis; the breakdown of RBC proteins; immunoevasion; enzyme activation; virulence; tissue and cellular invasion; and excystment, hatching, and moulting [25,26]. These functional diversities are being contributed to by their unique nucleophilicity, adaptability to different substrates, stability in different biological environments, and regulation [27]. Due to their critical functions in many parasites’ life cycles and pathogenicity, these cysteine proteases are found suitable to be the target molecules to combat *E. histolytica* growth by blocking interactions that cause these proteases’ inactivity. Research conducted by Tusar et al. (2021) [24] reported a few inhibitors that can target the cysteine protease: macrocypins, thyropins, and serpins. All three inhibitors can bind to two different cysteine protease families together with their two reactive sites. However, cysteine proteases as therapeutic targets have not yet been brought into clinical trial [28].

### 2.3. Protein Phosphatases

Protein phosphatase plays a vital role in regulating any organism, including this parasite; thus, it is considered a potential therapeutic target. In *E. histolytica*, this peripheral membrane protein can also be found in several places such as the cytoplasm, plasma membrane, chloroplast, nuclear, and cytoplasmic [29]. A dedicated database and web server, namely ‘EhPPTome’, was developed by Anwar and Gourinath (2017) [29] which incorporates information about 250 protein phosphatases in *E. histolytica* that helps to improve the understanding of the background protein phosphatases. The EhPPTome includes the classification of phosphatases into the superfamily and families, their localization, their biological function, and their KEGG pathway. This understanding makes phosphatases a potential target for *E. histolytica*.

A study by Ehrenkaufer et al. (2018) [8] identified PP2a as a potential drug target for entamoeba due to its important functions and targeting of calyculin, fostriecin, and okadaic acid. Besides, research conducted by Tapia et al. (2015) [30] also explained the protein tyrosine phosphatase of regenerating liver (PRL), a group of phosphatases that has not been broadly studied in protozoan parasites. This enzymatic activity can be inhibited by the PTP inhibitor o-vanadate when the recombinant protein (rEhPRL) has an enzymatic activity with the 3-o-methyl fluorescein phosphate (OMFP) substrate. The results show that the enzymatic activity was completely inhibited when assayed using 1, 3, and 5 mM for 2 h. Further characterization of the o-vanadate effects by a dose-dependent inhibition assay with concentrations <1 mM produces an IC50 value of 0.169 mM [30].

The recent research regarding protein phosphatases is the introduction of new low molecular weight tyrosine phosphatases encoded by EhLMW-PTPs genes [31]. This protein was found to have an essential role in the proliferation, the adhesion, the phagocytosis mechanisms, and regulating the pathogenicity of *E. histolytica*. Targeting of this protein was conducted by Sierra-Lopez et al. (2021) [31] using orthovanadate, which shows the inhibition of enzymatic activity.

### 2.4. Triosephosphate Isomerase

Triosephosphate isomerase (TIM) is one of the proteins indirectly associated with the plasma membrane of *E. histolytica* [16].This protein is involved in the first step of the subpathway that synthesizes D-glyceraldehyde 3-phosphate from the glycerone phosphate of glucogenesis. A study by Vique-Sanchez et al. (2021) [32] found that the 5,5′-[(4-nitrophenyl)methylene] bis(6-hydroxy-2-mercapto-3-methyl-4(3H)-pyrimidinone; known as D4) against the triosephosphate isomerase of *Trichomonas vaginalis* (TvTIM) has an amoebicidal effect on in vitro cultures, with an IC50 value of 18.5 µM. They also tested the D4 to the Triosephosphate *E. histolytica*. Results show that compound D4 targeting specific interaction sites Lys77, His110, Gln115, and Glu118 has favorable experimental and theoretical toxicity. Therefore, D4 should be further investigated as a potential drug against *E. histolytica*.

### 2.5. Alcohol Dehydrogenase

Alcohol dehydrogenase (ADH) is a type of enzyme found in two places in *E. histolytica*, which are in the cytoplasm and at the peripheral surface of the amoeba’s trophozoites [16]. This enzyme is crucial in the amoebic fermentation pathway as it helps to metabolize the ethanol to produce energy, notably in the *E. histolytica*, as it lacks energy-producing mitochondria [33,34]. According to Konig et al. (2020) [35], EhADH1 families are localized in the cytoplasm. Meanwhile, the EhADH2 and EhADH3 family members are located on the cell surface of amoebae and secreted or shed extracellularly. The *E. histolyica’s* alcohol dehydrogenase 2 (EhADH2) is a bifunctional enzyme with aldehyde dehydrogenase (ALDH) and ADH activities, which are essential for the growth and survival of an organism [34].

Furthermore, these alcohol dehydrogenase-E enzymes (ADHE) are present in various organisms and become an important target for anti-microbial agents. Furthermore, phylogenetic tools have suggested that EhADH2 is similar to other protist and bacterial bifunctional enzymes [34]. Therefore, identifying inhibitory sites with enzyme inhibitor databases is helpful for EhADH2 and ADHE homologs by using efficient lab-tested pyrazoline derivatives, which can inhibit enzyme activities and parasite growth [34].

### 2.6. GTPases

GTPases are molecular switches that regulate cellular processes such as cell polarity, gene transcription, microtubule dynamics, the cell cycle, cell migration, and vesicle trafficking [9,36]. A recent study by Verma et al. (2020) [37] characterized a Ca^2+^-binding protein from *E. histolytica* (EhCaBP6) as a novel GTPase. EhCaBP6 can be localized in the nucleus, cytoplasm, and plasma membrane, and are sensitive to heat stress. Increased expression of EhCaBP6 correlated with a significant increase in the number of microtubule structures, indicating that this protein can regulate chromosomal segregation and cytokinesis in *E. histolytica*. Downregulation of EhCaBP6 affected cell proliferation by causing delays in the transition from the G1 to S phase and inhibition of both DNA synthesis and cytokinesis. Their research also demonstrated that EhCaBP6 modulates the dynamic of microtubules by increasing the rate of tubulin polymerization.

Furthermore, their results show that EhCaBP6 is an unusual CaBP involved in regulating cell proliferation in *E. histolytica*, similar to nuclear Calmodulin (CaM). Thus, targeting CaBP gives promising results in inhibiting the growth of *E. histolytica*.

The Rho family GTPase is the other GTPase that can be the drug target molecule [38]. These proteins have different isoforms and were found to increase and decrease the membrane fractions in an abundance. The increased abundance of GTPase proteins may give a possibility of effective colonization and invasion of the trophozoites in the host. The Rho family GTPase’s proteins play essential roles in the virulence of the parasite with other identified proteins such as the Gal/GalNAc subunit, NAD. (P) transhydrogenase alpha, and calreticulin [13]. The Rho family GTPase also was found in abundance in the membrane fraction of *E. histolytica* when a study was conducted by Azmi and Othman (2021) [13]. The significant interactions between these proteins would give an understanding of the pathogenesis of the *E. histolytica*. Thus, further study needs to confirm their localization and functions. Therefore, targeting *E. histolytica* GTPase with compounds may inhibit its virulence.

### 2.7. KERP1

The KERP1 protein, which is rich in lysine and glutamic acid, is a virulence factor expressed on the cell surface in the human pathogen *E. histolytica* [39]. Microscopic data showed that KERP1 accumulated in vesicles of different morphologies that appeared to move randomly. They depended on the actin-rich cytoskeleton but were independent of the antegrade transport. The protein has been associated with endomembrane transport components such as multivesicular endosomes/bodies (MVB) and specific phosphatidylinositols (PtdIns). The subcellular localization of KERP1 compared to known markers for vesicle transport showed the intracellular transit of KERP1 as a cargo molecule and during interaction with host pathogens from the parasite to the external environment. Furthermore, this protein participates in adhesion and cytotoxicity when it comes into contact with human enterocytes or sinusoidal endothelial cells of the liver [40]. Apart from that, downregulation of the KERP1 protein level in virulent trophozoites leads to inhibition of liver abscess formation in animal models [41]. However, the studies related to the drug targeting of this protein are still unknown, although it possesses a significant function in the pathogenesis of *E. histolytica*.

### 2.8. Protein Kinase

These protein kinases are known to regulate multiple cellular processes such as metabolism, motility, and endocytosis through the phosphorylation of specific target proteins that form a communication system that sends extracellular signals for an adaptive response to the intracellular environment. Furthermore, the kinome of the parasite is composed of several conserved kinases with an unusual domain architecture. About one-third of kinome codes for transmembrane kinases (TMK) is proposed to help the parasite sense and adapt to the gut environment, which is constantly changing.

The abundant number of kinases that *E. histolytica* possesses allows us to assume that the regulation of cellular functions by phosphorylation/dephosphorylation processes is critical. The genome of this parasite codes for 331 putative protein kinases, which account for 3.7% of the proteome [42]. Protein kinases are the second most important drug target group after the G protein-coupled receptor due to their function in parasite proliferation and invasive disease formation [43].

*E. histolytica* contains a large and novel family of transmembrane kinases (TMKs), specifically approximately 100 putative transmembrane kinases (TMKs), indicating that the parasite has extensive means of environmental sensing [44,45]. According to Christy and Petri (2014), the transmembrane kinases (TMKs) of *E. histolytica* are a recently discovered family of cell surface proteins that share sequence similarities with the intermediate Gal/GalNAc-adhering lectin subunit.

TMKs are become potential targets for drug development due to their demand for virulence and proliferation. A study conducted by Abhyankar et al. (2012) [44] further confirmed the role of EhTMKB1-9 as an important virulence factor and opened up possibilities of screening molecules that potentially could target this and other similar extracellular receptors. A similar observation has been pursued by Lopez- Contretras et al. (2017) [46] in which Src kinases of *E. histolytica* are a critical factor in the biology of this parasite through the regulation of actin cytoskeletal remodeling via the activation of RhoA GTPase. Therefore, it could be a target molecule for the design of future drugs capable of preventing disease transmission.

Recent studies conducted by Sauvey et al. (2021) [47] tested a set of twelve FDA-approved antineoplastic kinase inhibitors against *E. histolytica* trophozoites in vitro. They identified dasatinib, bosutinib, and ibrutinib as amoebicidal agents at low molecular concentrations. Furthermore, twelve additional drugs developed from the computational tool found that ponatinib, neratinib, and olmutinib are highly potent, with EC50 values in the sub-micromolar range. Six drugs were found to kill *E. histolytica* trophozoites as rapidly as metronidazole. These findings revealed antineoplastic kinase inhibitors as an up-and-coming class of potent drugs against this widespread and devastating disease. Kinase research was also conducted by testing with the adenosine 5′-phosphosulfate kinase (EhAPSK). Validation was conducted using a combination approach of in silico molecular docking analysis and an in vitro enzyme activity assay for large-scale screening. Compounds 2-(3-fluorophenoxy)-N-[4-(2-pyridyl)thiazol-2-yl]-acetamide, 3-phenyl-N-[4-(2-pyridyl)thiazol-2-yl]-imidazole-4-carboxamide, and auranofin were identified as EhAPSK inhibitors [48].

### 2.9. Nickman Pick Type (NPC)

Research on lipid synthesis and metabolic pathways may be a promising area for developing effective vaccines and antiparasitic drugs. Of these, cholesterol is a fundamental molecule for the expression of virulence that enhances the trophozoite’s adherence to the host cells and extracellular matrix [49]. However, the trophozoites of *E. histolytica* lack enzymes for cholesterol and fatty acids synthesis, which they need to scavenge from the host or culture medium by specific mechanisms. A study conducted by Bolanos et al. (2017) [50] investigated the presence of NPC1 and NPC2 proteins in the trophozoites of the parasite, wherein both proteins are involved in the cholesterol trafficking in mammals. Bioinformatics analysis revealed one Ehnpc1 and two Ehnpc2 genes expressed as a transmembrane protein and EhNPC2 as a peripheral membrane protein of the trophozoites. These proteins were found to play a key role in cholesterol uptake. Targeting both proteins may probably disrupt the trafficking of cholesterol, which is needed by the *E. histolytica* to express its virulence. However, no current study is being performed to target these proteins.

### 2.10. Interferon-Gamma (IFN-γ) Receptor

Martinez-Hernandez et al. (2019) [51] identified IFN-γ receptor-like proteins on the surface of *E. histolytica* trophozoites using anti-IFN-γ receptor 1 (IFN-γR1) antibody through several techniques such as immunofluorescence, western blot, protein sequencing, and in silico analyses. The protein was found to modulate parasite virulence. The surface IFNγ receptor-like protein is functional and may play a role in disease pathogenesis and/or immune evasion [52]. The coupling of human IFN-γ to the IFN-γ receptor-like protein on live *E. histolytica* trophozoites had significantly upregulated the expression of *E. histolytica* cysteine protease A1 (EhCP-A1), EhCP-A2, EhCP-A4, EhCP-A5, amebapore A (APA), cyclooxygenase 1 (Cox-1), Gal-lectin (Hgl), and peroxiredoxin (Prx) in a time-dependent manner [52]. Furthermore, IFNγ signaling through the IFNγ receptor-like protein increased the erythrophagocytosis of human red blood cells of *E. histolytica*. However, this signaling was abolished by the STAT1 inhibitor fludarabine. In addition, the exogenous IFNγ potentiates *E. histolytica* chemotaxis, kills colon carcinoma (Caco2) and Hep G2 colonic liver cells, and increases amoebic liver abscess (ALA) formation. Thus, targeting this receptor protein could disrupt the pathway and combat the *E. histolytica*.

### 2.11. ERGIC53-like Protein

The ERGIC53-like protein is a type I membrane protein. It belongs to the class of intracellular cargo receptors and is essential for intracellular glycoprotein transport. They are involved in the transport of glycoproteins between the endoplasmic reticulum and the Golgi body. According to Khan and Suguna (2019) [53], the structure of the domain bears a resemblance to mammalian and yeast orthologs ERGIC-53 and Emp46, respectively. However, there are significant changes in the carbohydrate-binding site. The difference that the ERGIC53 protein of *E. histolytica* portrays is that it may probably have potential as a drug target.

## 3. Potential Compound

Potential compounds are compounds that are shown to have anti-amoebic activity. It can be divided into two groups, which are natural and synthetic compounds. These compounds were experimentally tested and resulted in inhibitions of the *E. histolytica.* However, the specific mode of action and the targeted compound are not being described clearly. Most of the compounds are still in studies and are not yet produced commercially.

### 3.1. Natural Products

Plants are a popular choice in developing countries as they can be considered safe and available at a low cost. The study conducted by Mehdi et al. (2019) [54] had shown experiments using *Tamarindus indica*. The *T. indica* extract in ethanolic and aqueous form was used against infected rats with *E. histolytica*. The *T. indica* contains many functional compounds such as flavonoids, alkaloids, tannins, phenols, and many more, making it suitable to kill *E. histolytica*. The alkaloids present in the plant extract had broken down the cell membrane of the *E. histolytica*, which causes the cell contents, such as the proteins and fat, to excrete. Besides, it also interferes with the DNA of *E. histolytica*. Both effects of the alkaloids result in the death of the *E. histolytica*.

Furthermore, tannins, another compound present in *T. indica* extract, inhibit the transport of proteins and enzymes on the cell membranes. The enzymes’ inhibition mechanisms are through the collaboration of tannins and the phenols, which results in protein precipitation through the formation of hydrogen bonds between hydroxyl phenols, nitrogen compounds, and proteins. The results showed the reduced number of *E. histolytica* in vivo when the dose of 500 mg/kg of the extract was used during the treatment of the rats. Apart from that, the extract also helps to repair the intestine tissue. This event can be seen from the histopathological section of the colon of the rats, where a moderate increase in the number of goblet cells and thickening of the mucosa of the colon when administered with *T. indica* extracts occur [54]. Increased goblet cells indicated that the immunity increased in the mucosa of the colon as well as the production of anti-microbial antibodies. Both events showed that this extract is a good indicator for curing patients with *E. histolytica* infection.

Another natural product used against *E. histolytica* is a *Camellia sinensis* extract. A study conducted by Shaker et al. (2018) [55] showed that *C. sinensis* extract is suitable for targeting the *E. histolytica*. It contains many beneficial compounds such as alkaloids, phenols, tannins, flavonoids, glycosides, saponins, and resins that kill *E. histolytica*. Both natural products are effective against *E. histolytica*. However, the specific biological membranes involved are not reported, which requires a deeper understanding of the mechanism of action.

### 3.2. Synthetic Compounds

Synthetic drugs are chemically produced in the laboratory and their chemical structures can be different or identical to naturally occurring drugs [56]. Research conducted by Inam et al. (2016) [57] had designed and synthesized a series of hydrazone hybrids (H1 Single bond H30) targeting *E. histolytica*. They found that the synthesized compound *N*′-(2-chlorobenzylidene)-4-(2-(dimethylamino) ethoxy) benzohydrazide exhibited promising results against *E. histolytica*. However, a specific mechanism of action was not published in detail. The compilation of the compounds and target proteins can be found in Table 1.

## 4. Synergistic Studies of Auranofin and Metronidazole

In Greek words, synergism comes from ‘synergos’, which means working together. More specifically, it refers to the interaction between two or more antibiotics of which the combined effect of both produces much more remarkable results than the sum of the antibiotics. Auranofin is a novel drug initially acting as an antirheumatic, which has been relabeled for the treatment of parasitic infections. Meanwhile, metronidazole is the most often prescribed drug. Both drugs have known potential in treating parasitic infection effectively. Due to their known potential, dual antibiotic therapy of both became the attention of some researchers as no dual antibiotic treatment of these drugs targeting *E. histolytica* had yet been conducted [58].

However, Owings et al. (2016) [59] conducted this dual antibiotic therapy targeting *Helicobacter pylori* and additive effects were produced. Addictive effects mean the combined effect of two antibiotics equal to the sum of the effect of antibiotics alone. Still, the results obtained from their experiments cannot directly determine the results of targeting *E. histolytica* using this dual antibiotic therapy. The reasons for this are because different species were targeted and probably different growth media was used. Various growth media influence the drug susceptibility as the compound would be different and affect the drug’s effectiveness. For example, cysteine, which is added in large amounts as an antioxidant, causes an increase of the toxicity of metronidazole [60].

Next, in a research study, Shaulov et al. (2021) [58] suggested that dual therapy of both antibiotics should be conducted. The experimental results showed that the adaptation of *E. histolytica* to auranofin results in metronidazole sensitivity. Thus, it can be concluded that dual therapy of these two drugs may be worth trying as no negative results were produced and it has the possibility of resulting in synergistic effects.

## 5. Conclusions

In conclusion, this review has discussed studies regarding the *E. histolytica* biological membranes, which have already been used as a compound or drug target. In future investigations, it might be possible to explore the potential of some discussed membrane and cytosolic proteins as potential drug targets to eliminate this protozoan parasite. Such potential proteins include thioredoxin reductase, cysteine protease, protein phosphatase, alcohol dehydrogenase, triosephosphate isomerase, GTPases, KERP1, the Nickman Pick Type (NPC), and the Interferon-gamma (IFN-γ) receptor. Targeting these proteins may help combat the *E. histolytica* as they are involved in its pathogenicity. Further research may use this paper to facilitate studies in obtaining potential molecules in drugs targeting *E. histolytica* and further characterize potential compounds. Dual antibiotic therapy of metronidazole and auranofin also can be considered for further study.

## Figures and Tables

**Figure 1 membranes-12-00396-f001:**
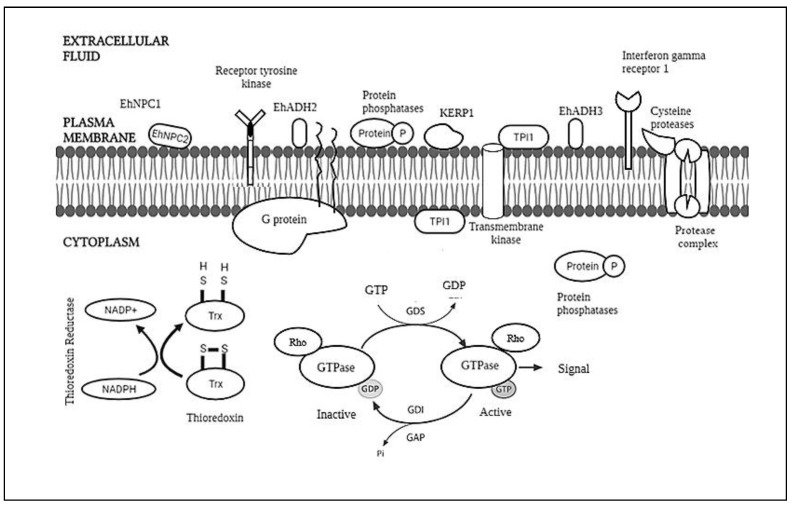
This is a figure of the membrane and its components.

**Table 1 membranes-12-00396-t001:** The table below summarizes the compilations of the compounds and target protein.

Reference	Protein Target	Compound
[22]	Thioredoxin reductase	Auranofin
[21]	Thioredoxin reductase	(1-(carboxymethyl)-4-(4-methylthiazole-5-carboxamido)-3H-pyrazol-1-ium-3-ide)
[23]	Thioredoxin reductase	*Homalomena aromatica* Schott 3,7-dimethylocta-1,6-dien-3-yl acetate, α-methyl-α-(4-methyl-3-pentenyl)-oriranemethanol 7-octadiene-2,6-diol-2,6-dimethyl
[24]	Cysteine protease	Macrocypins, thyropins, and serpins
[8]	Protein phosphatases2 a	Calyculin, fostriecin, and okadaic acid
[30]	Recombinant tyrosine phosphatase regenerating liver	P.T.P. inhibitor o-vanadate
[31]	Low molecular weight tyrosine phosphatases	Orthovanadate
[32]	Triosephosphate isomerase	5,5′-[(4-nitrophenyl) methylene] bis(6-hydroxy-2-mercapto-3-methyl-4(3H)-pyrimidinone) or D4
[34]	Alcohol dehydrogenase	Lab-tested pyrazoline derivatives
[37]	EhCaBP6	-
[13]	Rho family GTPases	-
[44]	KERP1	-
[47]	Kinase	Dasatinib, bosutinib, ibrutinib, ponatinib, neratinib, and olmutinib
[44]	EhTMKB1-9	-
[46]	Src kinases	-
[48]	Adenosine 5′-phosphosulfate kinase (EhAPSK)	2-(3-fluorophenoxy)- N-[4-(2-pyridyl)thiazol-2-yl] -acetamide, 3-phenyl-N-[4-(2-pyridyl)thiazol-2-yl]-imidazole-4-carboxamide
[48]	Adenosine 5′-phosphosulfate kinase (EhAPSK)	Auranofin
[38]	Nickman Pick Type 1	-
[38]	Nickman Pick Type 2	-
[51]	Interferon-gamma (IFN-γ) receptor	STAT1 inhibitor fludarabine
[53]	ERGIC53-like protein	-
[54]	-	*Tamarindus indica* flavonoids alkaloids tannins phenols
[55]	-	*Camellia sinensis.* alkaloids, phenols, tannins, flavonoids, glycosides, saponins, resins
[57]	-	*N*′-(2-chlorobenzylidene)-4-(2-(dimethylamino) ethoxy) benzohydrazide

## Data Availability

Not applicable.

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
