# Peer review of "A Review: Natural and Synthetic Compounds Targeting Entamoeba histolytica and Its Biological Membrane"

_membranes, 2022, doi:10.3390/membranes12040396_

Round 1

Reviewer 1 Report

This manuscript "A review: Natural and synthetic compounds targeting E.
histolytica biological membrane" is well written and very interesting because it describes in detail both natural and synthetic compounds E. histolytica biological membrane in the period from 2015-2021. The review is very important because amoebiasis is the third most common parasitic cause of morbidity and mortality, particularly in countries with poor hygienic settings in central and south America, Africa, and India. This disease is caused by Entamoeba histolytica, which infects approximately 50 million people worldwide, resulting in 70,000 deaths every year.

The aim of this review is to describe all known components of E. histolytica the biological membrane, which may help further research regarding the drug targeting of these molecules.

My suggestion is to add a schematic presentation of the membrane E. histolytica with components.

Check all references which must be cited in the same way (13, 58).

Author Response

Dear Reviewer,

Thanks for the positive comment. We have amend the manuscript accordingly.Please see the attach file.

Corresponding author,

Dr.Nurulhasanah Othman

Reviewer 2 Report

this review is a useful summary of proteins embedded in the plasma membrane of Entamoeba histolytica. It does summarise natural and synthetic compounds but this comes at the end of the paper and not all are reported to target the membrane. Perhaps the title of this submission should be changed to reflect this. I think that the content is broader than the title specified. 

Elements of the paper should be further developed for a further critical appraisal of future developments. For example what does metranidazole target. what are the implications? Are synergistic studies worth considering? Auranofin is a commercially available compound. Can synergistic studies be considered? 

This would push the review from a simple summary to a detailed reflective piece on drug developments in this field

Author Response

Dear Reviewer,

Thanks for the constructive comments. Its really help us in improving the write-up of the manuscript. Please see the attach file for details of the revised version.

Round 2

Reviewer 1 Report

The authors successfully added the schematic diagram of the E. histolytica membrane and my opinion is that the manuscript is suitable for publication in Membranes.